# Needle in a haystack: Harnessing AI in drug patent searches and prediction

**Leonardo Costa Ribeiro**[1], **Valbona Muzaka** [2]*

**1** Departamento de Ciências Econômicas, Faculdade de Ciências Econômicas, Universidade Federal de Minas Gerais, Belo Horizonte, Minas Gerais, Brasil, **2** Economic-History Department, Uppsala University, Uppsala, Sweden

☯ All these authors are contributed equally to this work.

* valbona.muzaka@ekhist.uu.se

## Abstract

The classification codes granted by patent offices are useful instruments for simplifying the bewildering variety of patents in existence. They are singularly unhelpful, however, in locating a specific subgroup of patents such as that of drug-related pharmaceutical patents for which no classification codes exist. Taking advantage of advances in artificial intelligence and in natural language processing in particular, we offer a new method of identifying chemical drug-related patents in this article. The aim is primarily that of demonstrating how the proverbial needle in a haystack was identified, namely through leveraging the superb pattern-recognition abilities of the BERT (Bidirectional Encoder Representations from Transformers) algorithm. We build three different databases to train our algorithm and fine-tune its abilities to identify the patent group in question by exposing it to additional texts containing structures that are much more likely to be present in them, until we obtain the highest possible F1-score, combined with an accuracy of 94.40%. We also demonstrate some possible uses of the algorithm. Its application to the US patent office database enables the identification of potential chemical drug patents up to ten years before drug approval, whereas its application to the German patent office reveals the regional nature of drug R&D and patenting strategies. The hope is that both the method proposed and its applications will be further refined and expanded forthwith.

## Introduction

Well before the Covid-19 pandemic was declared, several entities filed patent applications related to the spike protein of the novel SARS-CoV-2 virus shortly after its publication in January 2020. Among these were companies such as Moderna and BioNTech, that would go on to gain market authorisation for their mRNA vaccines later that year, and scientists such as Katalin Karikó and Drew Weisman who won the Nobel Prize for their work on mRNA in 2023. The multiple legal challenges over mRNA patents that have followed since testify not only to the tension inherent in granting patent rights to specific assignees for cumulative and collaborative work developed over time—mRNA was discovered in 1960 and scientists have been

**Data Availability Statement:** We created a number of datasets to train our algorithm, as detailed in the article. Most of the data is publicly available, but some restrictions apply to other kinds of data, as explained below. The Orange Book Database is publicly available and its URL is provided in the

article. The BERT algorithm is publicly available here: https://huggingface.co/docs/transformers/model_doc/bert The Biomedical DistilBERT is also publicly available here: https://huggingface.co/nlpie/distil-biobert To translate patent information from German to English we used Facebook's 667 state-of-the-art M2M-100 model, a non-English centric model which is available publicly at: https://huggingface.co/facebook/m2m100_418M General patent information from USPTO is also publicly available and can be found at: https://ppubs.uspto.gov/pubwebapp/static/pages/landing.html Patent information and related data pertaining to PATSTAT, DPMA (German Patent Office) and IQVIA is paywalled and not in the public domain. DPMA data was obtained from the PATSTAT database. This part of the data underlying the results presented in the study are available from these parties' websites, respectively at: PATSTAT: https://www.epo.org/en/searching-for-patents/business/patstat IQVIA: https://www.iqvia.com/.

**Funding:** Author who received the awards: LCR The work undertaken for this paper was financed in part by the Coordenação de Aperfeiçoamento de Pessoal de Nível Superior (CAPES), grant 88887.837596/2023-00 and by Conselho Nacional de Desenvolvimento Científico e Tecnológico (CNPq), grant 312020/2021-0 The funders did not play any role in the study design, data collection and analysis, decision to publish, or preparation of the manuscript. Funders websites: CAPES: https://www.gov.br/capes/pt-br CNPq: https://www.gov.br/cnpq/pt-br.

**Competing interests:** The authors have declared that no competing interests exist.

working on developing mRNA vaccines with lipid-based delivery systems since the late 1970s [1]—but also to the recent transformation of pharmaceutical patents into highly valuable assets. Indeed, considerable amounts of public funding behind the knowledge embodied in the successful Covid-19 vaccines notwithstanding, it is the companies controlling the crucial patents and other forms of intellectual property (IP) that have reaped the Covid-19 vaccines' financial benefits.

Partly because of the socio-economic importance of the products the *proprietary* pharmaceutical sector generates, and partly because of the important role patents have historically played in it, research on pharmaceutical patents is well-established in a number of disciplines, including philosophy, economics, business studies, law, and science and technology studies. However, scholars inclined to work with patents as their primary material—and users of the patent system at large—sooner or later encounter difficulties in identifying specific groups of pharmaceutical patents. One of the most of important patent groups, unsurprisingly for a sector focused on the development and introduction of new drugs, is that related to drugs. We use the term drug to include all medicines for human therapeutic use regardless of their type (e.g. chemical or biologic drug) and of their delivery method (e.g. pill, gel, injection etc.) Patent offices typically receive pharmaceutical patent applications related to a therapeutic agent well before it becomes embodied in a drug authorised for human use, and the attrition rate is high [2]. To be sure, such patents are categorised and indexed, but the problem persists for, as we go on to explain in more detail, there is no specific patent classification for pharmaceutical drugs. Moreover, although the rights associated with a pharmaceutical patent are national, the application process can follow national, regional and quasi-global routes (e.g. the International Patent Classification (IPC) system of the World Intellectual Property Organization, WIPO), thus resulting in a multiplicity of pharmaceutical patent codes, none of which is designated to pharmaceutical drugs. The distinct development trajectories of national patent laws and patent offices, the emergence of regional and international harmonisation initiatives, constant technological developments in the field, the rapid increase in pharmaceutical patent numbers worldwide, alongside insufficient attention to their indexing, have combined to make drug-related patents even harder to identify and therefore work with and analyse.

In the face of this mismatch between the importance of pharmaceutical drug patents—as highly valuable assets and as research objects—and their relatively poor searchability, this article proposes a method for facilitating their identification across the world. For reasons we come to below, the focus here is on small-molecule drugs derived from chemical synthesis (herein, chemical drugs). Biologic drugs are usually large, complex molecules derived from living cells or through biological processes and pose a different set of challenges that we are currently working to address separately. The mainstay of the sector for over 100 years, chemical drugs still account for the majority of all drugs sold globally [3], despite the promise of biologics in time overtaking them in the pharmaceutical market. We find, however, that chemical drug patents constitute a relatively small share of pharmaceutical patents overall—a needle in a haystack—a somewhat surprising outcome for a sector heavily reliant on patents and focused on the introduction of new drugs.

We start by explaining the nature of the identified mismatch in the following section, before proposing natural language processing, specifically, the BERT (Bidirectional Encoder Representations from Transformers) algorithm, as a way of rectifying it in the third one. The contribution this article makes is predominantly methodological, i.e. the introduction of a method for identifying a specific category of patents not captured by available patent classification codes. For this reason, we invest more time in explaining the BERT algorithm as a powerful natural language processing tool in section three, and the methodology we use to train it for the task at hand in section four. We obtain very promising results, namely a trained algorithm

with an overall performance (F1) score of 0.94. Despite our focus on methodology, we are also keen to demonstrate some of its possible uses, which is why, after discussing our results in section five, we use the trained algorithm to identify chemical drug patents in two patent offices in the final section. Our method is inspired not by faith in AI tools like BERT possessing some form of superhuman cognition, but by their usefulness as knowledge instruments designed by us to address an identified problem. In explaining our model in detail we engage, inevitably, in a form of 'demystification', driven in part by a commitment to keep AI tools open.

## Pharmaceutical drug patents in principle and practice

Couched predominantly in philosophical and legal terms, justifications for why patents are granted at all can be disarticulated from how patents operate in practice. Today, most pharmaceutical patents are used and managed as intangible assets, but the justifications for their grant and enforcement through public law are the same as during the 19th century. As discussed in a seminal article by Machlup and Penrose [4], these were/are that: inventors have natural property rights over their inventive ideas; patent rights constitute a just reward to inventors; patents provide the necessary incentives for innovation; and, society grants time-limited monopoly rights to inventors in exchange for the latter making public their invention. The latter justification is central to the existence of public, searchable databases listing patents issued in all fields of technology, whereas the others are often merged in the utilitarian argument that (pharmaceutical) patents provide the necessary financial incentives—via higher profits—to develop new drugs that would otherwise not be developed due to increasingly complex and costly nature of pharmaceutical R&D. Since Mansfield's classical study in 1986 [5], hardly anyone disagrees that the pharmaceutical sector is particularly reliant on patents due to a combination of relatively high costs for developing new drugs and their relatively high vulnerability to copying. Indeed, the sector stands alone in the extent of its involvement with the patent system and has done much to ensure that this system meets its requirements [6, 7].

A well-documented case of such involvement is the making of the 1994 WTO TRIPS (Trade-related Aspects of Intellectual Property Rights) Agreement which, as sector representatives were keen to boast, met most of the proprietary pharmaceutical sector's demands [8]. The number of *annual* pharmaceutical patent applications worldwide grew rapidly from under 20,000 in 1980 to around 100,000 in 2021, based on PATSTAT data, using IPC codes C07D, C07C, A61P and A61K. At the same time, the link between the justification of patents on account of their innovation-stimulating effects and actual innovation appeared to have broken: the number of new drugs introduced annually in the market decreased from an average of 93 in the 1960s to 31 in the 1996-2014 period [9]. In the US, for instance, nearly 80% of the drugs associated with new patents in the 2000-2015 period were existing drugs in the FDA (Food and Drug Administration) records [10].

There are numerous explanations for the rapid increase in pharmaceutical patent numbers without the corresponding rise in R&D productivity. The practice of 'evergreening', extending the life of a drug patent by obtaining additional protections for minor modifications or additions, alongside that of 'patent fencing', obtaining many patent families covering various aspects of a single drug, are both well-known in the sector (e.g. [11, 12]), and help explain the multiplication of pharmaceutical patent numbers with no corresponding rise in R&D productivity. A number of accounts have instead focused on the knowledge-specific difficulties associated with biopharmaceutical R&D processes to explain the phenomenon of disappointing R&D productivity (e.g. [13, 14]). Most accounts, however, tend to neglect broader structural changes that have occurred in the sector over the last three decades or so. Importantly, the core activities of large proprietary companies have shifted away from in-house R&D processes

towards primarily leveraging knowledge in and through networks where much of research, development, and production activities now take place [15]. This new way of operating is based on large companies' control over and management of intangible assets of all kinds, of which IPRs in general and patents in particular are a significant category [16]. Pharmaceutical patents—conceptualised both as legal entitlements and as material manifestations of future earning streams—are now largely constructed, bundled, and managed as assets critical to rent extraction in the sector (e.g. [15, 17, 18]. This shift towards the control and management of intangible assets is visible in large companies' balance sheets: the ratio of intangible over total assets of the largest 27 proprietary pharmaceutical companies (controlling over 2/3 of the global pharmaceutical market) increased from an average of 13 percent in 2000 to 51 percent in 2018 [19]. This new way of operating remains very profitable for the proprietary pharmaceutical companies despite them introducing relatively fewer new drugs in the market [20, 21].

The disarticulation between patents' *raison d'être* and their current instantiation is at the heart of the mismatch we aim to rectify: the more pharmaceutical patents are managed as highly valuable assets, the higher the need of groups who work with them—researchers of various kinds, competitors, the legal profession, lawmakers, policymakers, civil society groups, society at large—to easily identify them. No difficulties exist in identifying pharmaceutical patents in general, using codes furnished by the IPC (e.g. A61K, A61K, C07K and C12N). But the classification categories do not enable the identification of specific pharmaceutical patent groups such as that of drug-related patents. A patent application related to a therapeutic agent is typically made before clinical trials for a treatment begin, and many such trials do not result in a drug approved for human use by the relevant authorities. A preliminary patent classification is first provided by the applicant and the relevant patent office then assigns classification numbers on the basis of patent legal claims so as to ease examiners' prior art search. More generally, classification codes granted by patent offices, despite their continues updates in response to technological changes, represent 'ideal' types that succeed in simplifying the bewildering variety of patents in existence. This simplification is useful for some purposes, but problematic for others, such as identifying the specific category of drug-related patents.

Precise classification is not merely a research problem. The pharmaceutical drug patent specifications are of crucial importance both at application/grant stage and at the enforcement stage; indeed, of the four patent justifications noted at the start, it is that of public disclosure that became over time the main one courts, especially in the Anglo-American tradition, came to rely upon [22, 23]. The legal literature on disclosure, for its part, laments that this important function of the patent system is limited in practice. Among other things, this is due to: the gaming of disclosure requirements to withhold commercially-valuable information; prioritising legal disclosure (patent claims) over technical disclosure so that the information contained is of limited practical use; the increasing complexity of legal patent claims aimed simultaneously at passing the originality test and expanding the outer limits of protected matter but without diminishing their defensibility in courts; and, importantly for our purposes, the difficulty and expense of locating patents in the vast database of issued patents (e.g. [22–26]). Not only is there insufficient attention paid to indexing patents and enhancing their searchability on the part of patent offices around the world; often enough, their classification can be arbitrary and therefore unreliable [24, 27]. In short, the opacity of patent texts, combined with their ever-increasing numbers and poor searchability, make patents uncertain entities for the majority of patent system users [23, 28, 29] precisely at a time when they have become crucial assets in the hands of large corporations.

There are four main reasons why identifying a sub-category of pharmaceutical drugs, such as drug-related ones, is worthwhile. First, developing new drugs is the core activity of the proprietary (as opposed to the generic) pharmaceutical sector. It is reasonable to assume that the

majority of pharmaceutical patents would be drug-related, but our analysis indicates that they constitute a minority of pharmaceutical patents in general. Given that large proprietary pharmaceutical companies are the main ones that are able to introduce new drugs in the market, and that their high valuation and profitability relies to a considerable extent on IPRs in general and patents in particular, this raises the question of what non-drug related patents are issued for and how they are used in the sector. Second, and related, a strong justification offered for pharmaceutical patents, both by the state and the sector, is that they are needed to undertake ever more complex and expensive R&D that results in new and more effective drugs in the market. The basis of this position in the face of historically lower new-to-the market drug introduction rates and a majority of non-drug-related patents would require, minimally, a re-evaluation. Third, the question of which category of pharmaceutical patents are the most valuable cannot be answered without being able to clearly identify different subcategories. As recent mRNA patent litigation has shown, due to a combination of high value, high technical complexity, and relatively high defensibility in courts, drug-related patents appear to be among the most hard-fought patent litigation cases [26, 30], highlighting again the need to identify specific subcategories of pharmaceutical patents. Fourth, many jurisdictions allow for pre- and/or post-patent grant challenges and quite a number have been successfully attempted not only by competitors and generic companies, but also by health-orientated civil society groups. The proper functioning of these kinds of provisions relies on fully-searchable, fine-indexed patent databases or, lacking these, as we indeed do, a reliable method of overcoming the difficulties that arise from relatively opaque and poorly indexed pharmaceutical patent data. These are the main reasons why we harness AI tools to enable the identification of drug-related patents.

We turn to AI and natural language processing (NLP) in particular not because we think it is a superhuman form of intelligence that can sort out the messy outcomes of historical contingencies in the field of pharmaceutical patents, but because it is a useful instrument that helps us secure better knowledge for our purposes. As we go on to show, NLP models and BERT in particular are very good at pattern recognition; their instrument-like nature is made clear in our account of BERT's training in which a considerable amount of 'craftmanship' is involved [31]. As with pharmaceutical patent databases, the way NLP models operate are often opaque, which is why we explain in detail how ours was built and trained. We start this process with the US patent office (USPTO); apart from the fact that the US constitutes the largest, most profitable, and hence the most attractive, pharmaceutical market in the world, the peculiar nature of the pharmaceutical patent application and drug approval processes there helps explain our chosen methodology, especially the training of the BERT algorithm.

Ours is not the first attempt to use AI tools in patent data analysis. Lee and colleagues [32], for instance, used an artificial neural network to analyse USPTO patents in order to identify emerging technologies. They achieved this by using 18 numerical indicators calculated from patent metadata, e.g. number of citations, the Herfindahl index on classes of cited patents, number of inventors, number of claims, and so on. These were used to train a neural network and generate two quantitative indicators that measure a technology's 'emergingness' over time. Besides not focusing on pharmaceutical patents, they made no use of patent texts at all, unlike our proposed method. Another notable example is offered by Balsmeier and colleagues [33] who made use of a NLP model to visualise co-inventor networks and patenting trends in the US during the 1976-2016 period. This was achieved through using NLP to analyse the names of investors, patent holders and their locations, not too dissimilar a method from that used in the commercial AI tools for patent searches that have become available recently. Once again, no engagement occurs with the claims and content of the patents analysed, as our model does.

## Methodology (I): Natural language processing

The most common way of searching patent databases—using keywords in the available search-able fields, namely the patent title and abstract—is also among the most ineffective in locating and identifying chemical drug patents. Even carefully selected keywords would miss many relevant patent documents. Keyword searches often do not allow for stemming—e.g. entering 'running' may not return documents containing 'run' or 'runner'—thus excluding potentially relevant findings from the list. More importantly, words often have multiple meanings that get determined by the surrounding context—take the example of 'bank', an institution, a river-bank, a storage location—thus leading to the opposite problem of including many irrelevant results. Difficulties related to language are compounded by technological change. Even if the most effective keywords have been identified, technological change may have rendered them ineffective search tools, a problem especially pronounced in emerging technological fields, such as AI or blockchain technology. Consider the field of AI, for instance, where ten years ago keywords like "machine learning", "deep learning", and "natural language processing" may have been sufficient to retrieve the majority of the relevant patents. As the field evolved, however, new techniques and associated terminology have emerged, such as "Reinforcement Learning", "Generative Adversarial Networks", and "Transformers".

Our aim is to provide a more sophisticated method of identifying all patents related to chemical drugs by using recently developed and advanced natural language processing tools. Although not addressing this question, the only related study we are aware of is a short feature by Chien and colleagues [34] who, responding to a request by USPTO for input on improving the quality of drug patents before they are granted, offer a predictive model for determining which pharmaceutical patents will likely become drug patents using machine learning based on numerical (i.e. not textual) data. Its methodology is based on computing and analysing statistics for various aspects of patent documents, e.g. families, citations, parties involved, and ownership. Unlike them, we rely on the pre-trained language model BERT and fine-tune it for the task in hand, i.e. we re-train it to enable the classification of patents drawn from large, general patent databases as belonging to the group of chemical drug patents. The relevance and applications of this method, as already noted and as we go on to explain in more detail in the last section, are much broader than the specific task set out by the USPTO.

Before explaining the steps taken to train the algorithm for these purposes, we justify in this section our choice of BERT through a brief account of its superiority over other language processing tools. A good place to start is Word Embedding, a technique used in NLP to represent words as numerical vectors in a high-dimensional space so that similar words are mapped to nearby points in it. The purpose is to capture the meaning and context of words in a way that can be understood by machines, enabling them to carry out various NLP tasks such as text classification, sentiment analysis, machine translation, and more. It addresses the problem afflicting more traditional techniques in which words would be represented as one-hot encodings that typically hinder machines' understanding of the text's meaning. Word embedding maps words to dense vectors in a high-dimensional space such that semantically similar words are clustered together. For example, words that have similar meanings, synonyms, antonyms, or words that appear in similar contexts, are mapped to nearby points in the vector space. This allows machines to capture the nuances of language and understand the relationships between words in a more human-like way.

Generally speaking, word embedding is a powerful technique for representing words as vectors, enabling machines to understand and process natural language more effectively. There are several techniques to learn word embeddings, including: (1) Word2Vec, a popular method

which uses two different techniques, namely continuous bag of words (CBOW) to predict a target word based on its surrounding context, and skip-gram to predict the surrounding context based on a target word; (2) Global Vectors for Word Representation (GloVe) which represents words as vectors in a high-dimensional space, based on their co-occurrence patterns in a corpus of text; and, (3) FastText, a method for learning word embeddings from raw text data, which uses a hierarchical approach to represent words at multiple scales. The performance of NLP tasks is greatly improved through word embedding. It can handle out-of-vocabulary words better than traditional methods through representing unseen words as vectors that are close to seen words with similar meanings. In short, it is generally better at capturing the relationship between words, subtle differences, and language nuances compared to traditional methods.

Despite these strengths, however, word embedding techniques are not as sensitive to the context in which a word appears as we would like. Due to its focus on the co-occurrence of words in a large corpus, word embedding is not as successful in capturing the different meanings of a word as used in a particular context. Going back to our earlier example of 'bank', its meaning would depend on the context in which it appears: 'I swam across the river to reach the bank' and 'I drove across the river to reach the bank' evidently refer to the riverbank and the financial institution respectively, with the verb determining the meaning in each instance. It is to address this lack of attention to context that attention-based models have been proposed. These models, often called Transformers, use attention mechanisms to focus on specific parts of the input text that are relevant and used them to transform the vector encoding of the other words in the sentence. These models enable some words (e.g. 'drove', 'swam') to alter the vector representation of others. Unlike recurrent neural networks, transformers are a type of neural network architecture that is particularly well-suited for sequence-to-sequence tasks, such as machine translation, summarisation, and text generation. This is because traditional recurrent neural networks process sequences one element at a time and use recurrent connections to capture long-term dependencies, whereas transformers process the entire input sequence in parallel and use self-attention mechanisms to weigh the importance of different parts of the input when generating the output. Furthermore, the attention mechanism is highly parallelisable compared to recurrent neural networks. It runs efficiently on a graphics processing unit and does not face the vanishing gradient problem that the long short-term memory neural network does.

Introduced initially by Vaswani et al. [35], transformers have become widely used in many NLP tasks. They consist of an encoder and a decoder, each composed of multiple identical layers. The encoder takes in a sequence of tokens (e.g., words embedding) and outputs a sequence of vectors, called "keys," "values", and "queries." The decoder then takes these vectors as input and generates an output sequence. The key innovation of transformers is the self-attention mechanism which, as noted, allows the model to attend to different parts of the input sequence simultaneously and generate output sequences of non-predefined length, capturing long-range dependencies and contextual information more effectively. Another important feature is their use of the positional encoding technique to preserve the order of the input sequence. Positional encoding adds a fixed vector to each input token, which encodes its position in the sequence. This allows the model to differentiate between tokens even if they have the same content. Because of these features, transformers have achieved state-of-the-art results on many NLP tasks, including machine translation, text generation, and question answering [36–38]. Importantly for our purposes, they have also been used as building blocks for other models, such as RoBERTa (Robustly Optimized BERT Pretraining Approach) and BERT (Bidirectional Encoder Representations from Transformers), which have achieved even more remarkable results on a wide range of NLP tasks.

## The BERT algorithm and its uses for text classification

Bidirectional Encoder Representations from Transformers, BERT for short, is a powerful NLP technique developed by Google in 2018. It is a pre-trained language model that uses a multi-layer bidirectional transformer encoder to generate contextualised representations of words in a sentence. These representations can be fine-tuned for a wide range of NLP tasks, such as sentiment analysis, question-answering, language translation and, importantly for our purposes, text classification. A brief account of BERT's architecture helps locate various features of the more advanced natural language processing techniques discussed so far, as well as provide the necessary background for explaining how we fine-tune it for our purposes. Its architecture consists of the following main components:

- BERT creates its own word embedding (vector representation of the words as Word2Vec and Glove) that is more sophisticated than the ones mentioned earlier. It uses a combination of token embeddings, segment embeddings, and positional embeddings to represent each token in the input. For words that are out of vocabulary, BERT word embedding splits them into subword tokens.

- Transformer Encoder: The input embeddings are fed into a multi-layer transformer encoder. Each layer in the encoder consists of two sub-layers: Self-Attention and Position-wise Feed-Forward Networks (FFN). As noted, the Self-Attention mechanism allows the model to weigh the importance of different words in the input sequence relative to each other and learn contextual relationships between them. The FFN processes the output of the Self-Attention layer to capture non-linear interactions between the input embeddings.

- Bidirectional Encoding: Unlike traditional language models that only consider the left or right context of a word, BERT uses a bidirectional approach that considers the entire input sequence in both forward and backward directions. This allows the model to capture both local and global dependencies between words.

- Pooler: The output of the transformer encoder is passed through a pooler layer, which aggregates the information from all positions in the sequence to form a fixed-length vector representation of the input text. This component can also include a neural network dense layer that performs the final classification, i.e., associates the transformer encoder with the class labels.

BERT has achieved a number of state-of-the-art results on numerous NLP benchmarks, including question answering (89,8% F1 score on the Stanford Question Answering Dataset [36]), named entity recognition (93.5% F1 score on the LINNEAUS dataset [37]), and text classification (98.9% accuracy on fake news detection [38]). But one of the most attractive features of BERT for our purposes is that it can be retrained to address new tasks while simultaneously leveraging the knowledge gained during the pre-training phase. Fine-tuning BERT to achieve more specific NLP tasks is achieved by inserting an additional layer on top of the pooler layer to adapt the model to the specific downstream NLP task. This layer is called a classifier. As we go on to show in the next section, the whole model needs to be retrained end-to-end using labelled data for the target task, in our case, classifying a pharmaceutical patent as a chemical drug patent i.e., a binary classification.

To fine-tune BERT for text classification purposes, the classifier layer added on top of the pooler layer is a neural network with softmax activation function. This layer will associate the vector encoding obtained in the pooler layer to a label that represents the class to which we want to classify the texts, thus providing the probability of the (patent) text belonging to a certain class, in our case that of a chemical drug patent. Although to the best of our knowledge

this is the first time BERT is fine-tuned for this purpose, interesting applications of BERT for text classification exist and include: (1) Spam vs. Non-Spam Emails (in one example, a team of researchers fine-tuned BERT on a dataset of 50,000 emails and achieved an F1 score of 0,969 [39]; (2) Medical Text Classification (such as identifying cancer types from clinical notes, out-performing other machine learning models in classifying breast cancer subtypes [40]; (3) Hate Speech Detection (in social media posts like Twitter, with a team achieving an F1 score of 0.91 through a fine-tuned BERT model [41]).

## Distillation in BERT

Distillation is a method used to transfer knowledge from a large, complex model (the "teacher" model) to a smaller, simpler model (the "student" model) while maintaining performance. In the case of BERT, distillation is used to condense the knowledge of a large, pre-trained BERT model (the teacher) into a smaller, more efficient model (the student). This process involves training the student model to emulate the behaviour of the teacher model using fewer parameters and computations [42]. During training, the student model aims to predict output probabilities similar to those of the teacher model by minimising the difference between their output distributions. This is achieved through a loss function called the knowledge distillation loss, typically a combination of cross-entropy loss and Kullback-Leibler divergence. By minimizing this loss, the student model is trained to match the output distributions of the teacher model. One popular example of BERT distillation is DistilBERT, which employs knowledge distillation during the pre-training phase to reduce the size of the BERT model by 40% while retaining 97% of its language understanding capabilities and achieving a 60% improvement in speed [42]. Once the student model is trained, it can be fine-tuned for specific tasks, such as sentiment analysis or question-answering, as the regular BERT algorithm.

## Methodology (II): Training the algorithm

### Creating the training sets

To fine-tune the BERT algorithm for specific classification purposes, as discussed, the addition of a classifier layer implemented via a neural network is needed for which, in turn, one or more training datasets are required. Ultimately with BERT, as with other NLP models, it is the quality of the training database that determines its accuracy. To train the algorithm to correctly classify chemical drug patents from the universe of pharmaceutical patents, building a pre-classified database containing chemical drug patents is a first crucial step.

Briefly put, the USPTO handles pharmaceutical patent applications and approvals and, as noted, they are submitted and granted years before there is a viable drug. Only a small part of this universe of drug-associated patents is of relevance when, following clinical trials, a new drug application is submitted for the purpose of gaining marketing approval by the US Food and Drug Administration (FDA). Under the 1984 Hatch-Waxman Act, new *chemical* drug sponsors, often large proprietary pharmaceutical companies, must list valid patents associated with the drug alongside their application to the FDA. Importantly, while the new drug sponsor has an obligation to provide and maintain accurate and complete drug patent information—e.g. patent number and its submission and expiry date for each of the patents related to specific aspects of the drug in the application—with the FDA, the role of the latter is purely ministerial, i.e. it is the responsibility of the new drug sponsor to determine and list relevant patents and their scope in the FDA application [26]. Should the new chemical drug be approved, the FDA merely publishes the list of patents related to the new drug as provided by the sponsor in the Approved Drug Products with Therapeutic Equivalence Evaluations, commonly known as the Orange Book [43]. The existence of the Orange Book is crucial to training our algorithm, since

it is chemical drug-related patents we want to be able to identify. The equivalent for biologic drugs, the so-called Purple Book [44], did not include such requirements and listing until 2020, this being a key reason why identifying biologic drug patents is relatively more difficult and why we address this challenge separately.

We obtained the FDA Orange Book December 2022 version comprising approximately 5,000 patents related to approved chemical drugs. The training set is by no means complete, however, because the Orange Book list the relevant chemical drug patents but it lacks metadata. To obtain these patents' metadata, we turn to PATSTAT, a patent statistics database provided by the European Patent Office (EPO) containing data on patent applications and grants from various countries, including those filed under the WIPO Patent Cooperation Treaty. It contains data such as patent titles, inventors, applicants, assignees, priority patents, filing dates, grant dates, and citations for more than 100 million patents from more than 90 countries, including major patent-granting authorities such as the USPTO, the Japanese Patent Office, and the Chinese National Intellectual Property Administration. We use the patent numbers of the nearly 5,000 chemical drug patents listed in the Orange Book to extract their title and abstracts and additional metadata from PATSTAT, thus creating the first database of known chemical drug patents. This part of the training dataset is illustrated as the step 1 in Fig 1. We opt to use patent abstracts to train the algorithm for two main reasons: first, they contain the main patent information in a shorter, summary form that helps reduce computational costs; and, second, as patent texts are much more likely to display local patenting idiosyncrasies, the use of more standardised abstract texts enhances the algorithm's applicability in patent offices around the world.

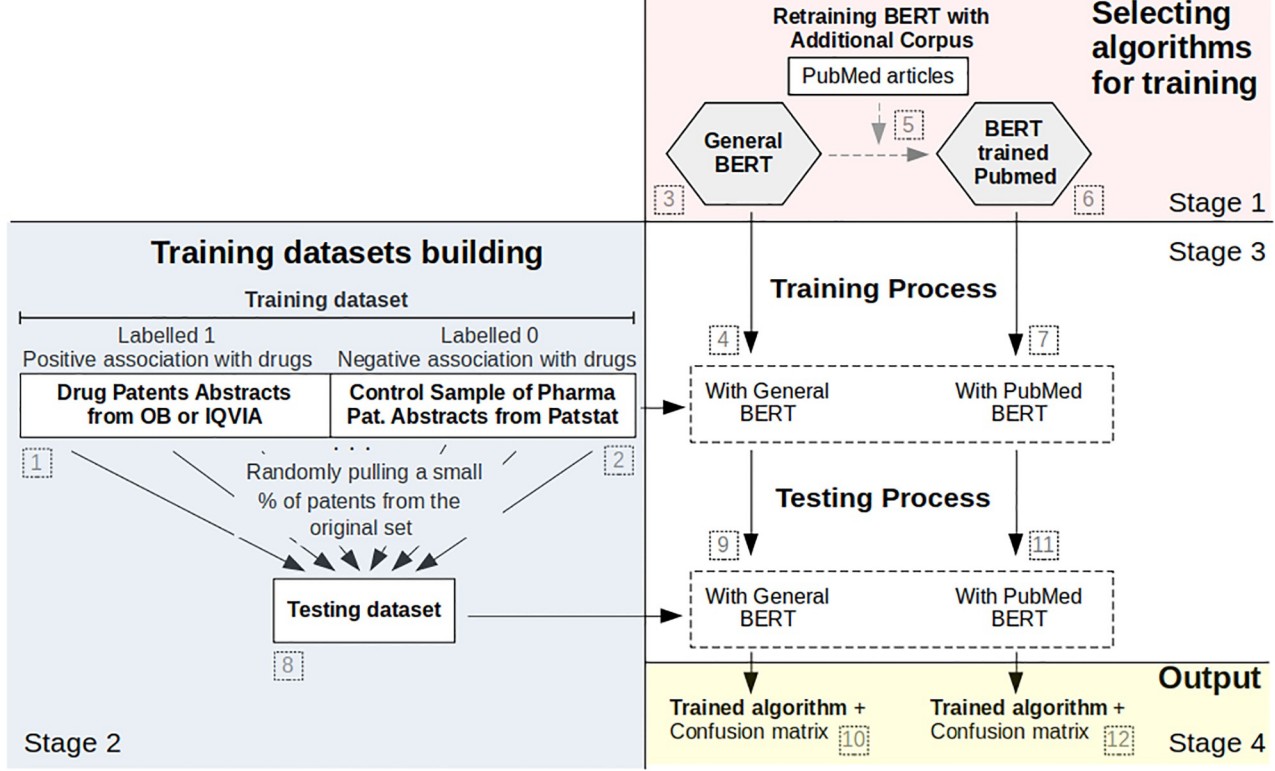

**Fig 1. Illustrative diagram of the algorithm selection, training dataset building and training/testing process.**

To train the algorithm comprehensively, a second database of a similar size is required. The first database includes documents we know are positively connected to chemical drugs and thus designated by the label 1. A second database would include pharmaceutical patents that lack explicit association with chemical drugs in the Orange Book, identified as 0 in our dataset. To compile this supplementary segment of the training data, we started with PATSTAT patents listed under pharmaceutical sector codes, but omitted the patents listed in the Orange Book thereby ensuring their affiliation with the pharmaceutical sector while excluding explicit association with chemical drugs. From this pool, we randomly selected approximately 5,000 patents, striking a balance between patents bearing labels 1 and 0 within our training dataset. This sample was obtained from a random selection using a uniform probability distribution among patents granted by the USPTO in the period from 1980 to 2023 (same temporal coverage as the Orange Book) and classified in IPC codes, C07K, C12N, A61K and A61O, the most commonly used codes in the pharmaceutical sector. This second part of the training dataset is illustrated as set 2 in Fig 1. Through this process, the first training containing around 10,000 patents was built—referred to simply as the Orange Book Patents dataset below—but this is not the only training dataset we considered in our analysis. Driven by the need to train and evaluate the training dataset that would generate the best possible accuracy after training the BERT algorithm, we created two additional training datasets in order to compare the precision results of correctly identifying patents associated with chemical drugs obtained from each of them.

The second training dataset was also built from chemical drug patents listed in the Orange Book, but in this case we supplemented the set with other patents from the relevant *patent families* obtained from PATSTAT. A patent family refers to a group of patents filed domestically and/or in different countries or regions that often share the same priority date and/or cover the same or similar technical content. There is no agreed definition of a patent family, but the most common types are equivalent, extended, and single-priority families [45]. Generally speaking, the patent family provides legal protection for an innovation across multiple jurisdictions, allowing the owner to prevent others from making, using, or selling the invention without permission in all of them. As patent families may contain the same patent applied for and granted in different offices, we included in this dataset only those that presented different abstracts to the ones we already had in the first training dataset so as avoid duplication. Around 8,000 patents associated directly or through their families with chemical drugs were obtained in this manner, which we labelled 1. This is also represented by the step 1 in Fig 1, but related to a different training dataset. As previously, we created a randomly selected dataset of an equal number of patents from the pharmaceutical sector that had not yet been labelled as 1. This part of the training dataset is illustrated as set 2 on the Fig 1. Therefore, our second training dataset—referred to as the 'Orange Book Patents & their Families' below—ended up having around 16,000 patents.

With an eye on improving accuracy, we built a third training dataset from a different source, namely IQVIA [46], a database that tracks thousands of drug patent families across 130 countries. Using IQVIA own classifications which tend to include a larger number of patent family members, we identified just over 10,000 patents associated with major chemical drugs until 2019. We find that only 36% of these patents are present in our Orange Book database, making the use of the IQVIA database to train and improve the performance of our algorithm worthwhile. Using their patent numbers, we returned to PATSTAT to obtain the relevant metadata, including these patents' abstracts. This is also represented by the step 1 on the Fig 1. As before, this training dataset was complemented by pairing this list of patents positively related to chemical drugs (labelled 1) with a set of around 10,000 patents randomly selected from PATSTAT that are classified as pharmaceutical sector patents but not yet identified as

chemical drug patents (represented by the step 2 in Fig 1). This third training dataset—corresponding to 'IQVIA Patents' in our discussion in section five—contained around 22,000 patents, i.e. it is over twice as large as the first 'Orange Book Patents' dataset discussed above.

## Training the algorithm

The training of the algorithm using the datasets built earlier was conducted using Python programming language and the open-source implementation of DistilBERT, discussed in more detail below. Fig 1, item 3, illustrates the chosen algorithm for training, and step 4 illustrates its training using different bases.

As noted earlier, one of the most attractive features of using BERT is that it can be retrained to address new tasks while simultaneously leveraging the knowledge gained during the pre-training phase. This ability to improve the performance of downstream tasks through leveraging previous knowledge is achieved through the so-called transfer learning mechanism, where the model uses the learned representations to adapt to new tasks with minimal additional training data. In short, the algorithm can be retrained with additional corpus, learning further specific semantic structures or meaning relations, adding these to what it had already learnt in previous training. It is this characteristic that Rohanian and colleagues [47] relied on to retrain the DistilBERT model using an additional corpus of articles published on PubMed. This retrained version of DistilBERT—herein Biomedical DistilBERT—is particularly relevant for us because of the existing synergies between the corpus used to retrain it and our focus on chemical drug patents. PubMed has little to say on the latter but, boasting a vast collection of publications related to life sciences and biomedical subjects drawn from the MEDLINE database, it provides an excellent additional corpus to train this version on DistilBERT on the terms and language structures specific to the domain of biomedicine and life sciences. It is reasonable to expect this more fine-tuned version to be better capable of handling terms and syntactic structures specific to domains closely related to the subject matter contained in chemical drug patents. This additional training is illustrated as the step 5 in Fig 1, providing the algorithm for training illustrated as item 6. In essence, we are increasing the models' capability to handle biomedical subjects to improve its accuracy in identifying our pharmaceutical patent category. We test whether accuracy is improved by using both Biomedical DistilBERT and the general (i.e. non-PubMed-trained) DistilBERT version, the latter referred to as 'General DistilBERT' in our discussion in the following section. The step 7 in Fig 1 corresponds to applying this fine-tuned version to our patent training dataset.

## Results: Assessing the identification performance

Aiming to train the BERT algorithm to correctly identify chemical drug patents from the universe of (pharmaceutical) patents, we built, as discussed, three different training patent datasets and used two versions of DistilBERT. These combinations enabled our AI model to be trained on a diverse range of patent information and enables us to evaluate their overall performance. To enable comparison between the different combinations/scenarios, we construct confusion matrices for each of them, namely (1) Orange Book Patents dataset and General DistilBERT Corpus combination; (2) Orange Book Patents dataset and Biomedical DistilBERT Corpus combination; (3) Orange Book Patents & their Families dataset and Biomedical DistilBERT Corpus combination; and (4) IQVIA Patents dataset and Biomedical DistilBERT Corpus one. All the results showed in the following section were obtained by running all combinations on a GPU with a batch size of 32, chosen due to memory limitations, and utilising 15 epochs due to the fact that the learning rate had already reached a significantly low value in all cases.

We use confusion matrices because they offer clear, tabular representations measuring the performance of a classification model. Specifically, they show the count of true positives (TP, correctly predicted positive cases), true negatives (TN, correctly predicted negative cases), false positives (FN, incorrectly predicted positive cases), and false negatives (FN, incorrectly predicted negative cases). Given our objective of identifying chemical patents, the true positive (TP) corresponds to the cells in the lower right of the matrices presented in this section. The false positive (FP) corresponds to the upper-right cell, the false negative (FN) to the lower-left cell, and the true negative (TN) to the upper-left cells. Importantly for our purposes, confusion matrices enable us to calculate several import metrics to access the performance of our proposed classification algorithm, namely:

- **Accuracy**—this measures the overall correctness of the model's predictions and is calculated as (TP+TN)/(TP+TN+FP+FN). It provides a general idea of how well the model is performing, but it may not be suitable when the class distribution is imbalanced.

- **Precision**—this measures the accuracy of positive predictions and is calculated as TP/(TP+FP). It indicates the proportion of positive predictions that were correct and is essential when minimising false positives is an important objective.

- **Recall/Sensitivity**—this measures the ability of the model to correctly identify all relevant positive instances and is calculated as TP/(TP+FN). It is crucial when avoiding missing positive instances is the main objective, even if it means including some false positives.

- **F1-Score**—this is the harmonic mean of Precision and Recall and is calculated as 2*(Precision*Recall)/(Precision+Recall). It is a particularly useful metric for evaluating overall performance that takes into account both false positives and false negatives. Because it provides a more comprehensive evaluation of the model's performance, we pay particular attention to the F1-score in the following analysis.

We start with the confusion matrix related to the scenario of using the Orange Book Patents training database discussed above and the General DistilBERT. Within the matrix, the integer values correspond to the number of patents correctly classified by the algorithm into each respective class, based on the testing dataset consisting of 1001 patents (10% of the training dataset, different percentages for dividing the training and testing sets were considered and the final decision was to use a 90%—10% separation because the model's accuracy plateaus above 90% of the training set, and the statistical fluctuation increases when the test set is relatively small). The testing dataset is a collection of data used to evaluate the performance of a machine learning model. The purpose of a testing dataset is to assess how well the model can generalise to new, unseen data after it has been trained on a separate training dataset. This testing dataset is illustrated as the item 8 in Fig 1. The vertical labels indicate the true classifications of the patents, while the horizontal labels represent the algorithm's predicted classes. When calculating accuracy using the values presented in Fig 2 (represented by the steps 9 and 10 in Fig 1), we obtain a result of 91.80%, i.e. this is the percentage of correct prediction rate for both positive and negative cases. The precision is 89.49%, representing the correct predictions within the positive class, and the recall is 93.66%, signifying the correct predictions within the category of actual drug patents. Finally, the F1-score, which provides an overall performance assessment of the classification, is 0.915. Misclassified patents were found upon further inspection to contain quite generic texts, this probably explaining the misclassification. False positives can in part be explained by the fact that there will be patents that are related to new drugs that have not yet completed the FDA approval process. These patents, while relevant, are not yet included in the Orange Book database and hence are not labelled as drug patents in our training dataset.

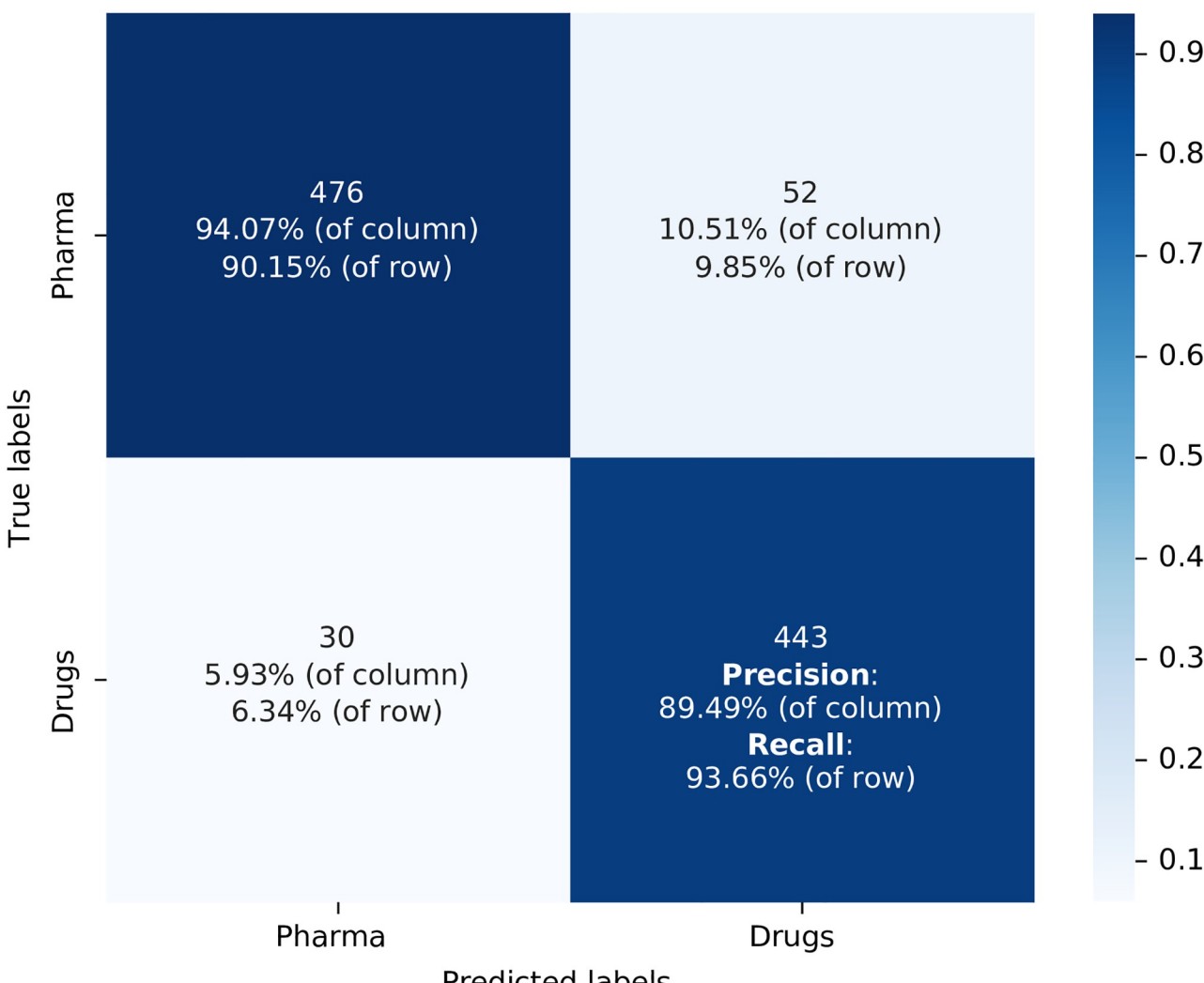

**Fig 2. Confusion matrix for the model used the training dataset built from Orange Book Patents with General DistilBert as the algorithm model label in Table 1 T: OB; C: G.** Source: Authors' elaboration

We build another confusion matrix, represented in Fig 3, for the scenario when the same training dataset as above was combined with Biomedical DistilBERT which, as discussed, was fine-tuned using the additional corpus of biomedical articles (represented by the steps 11 and 12 in Fig 1). Now, when we calculate accuracy, we achieve a 93.40% result, along with 91.62% precision, 94.71% recall, and an F1-score of 0.931. Comparing these two scenarios, it is evident that all performance metrics improved with the use of the Biomedical DistilBERT version. In line with our expectations, the fact that this version of DistilBERT has been fine-tuned with a corpus of material that overlaps with the subject matter of the patents we want to classify, has contributed to improved performance metrics. This is why we do not use the General Distil-BERT version in the remaining combinations.

The third combination involves the Orange Book Patents and their Families dataset and the Biomedical DistilBert Corpus. The confusion matrix for this model is displayed in Fig 4 (represented by the steps 11 and 10 in Fig 1). To compute the confusion matrix, we utilised a testing

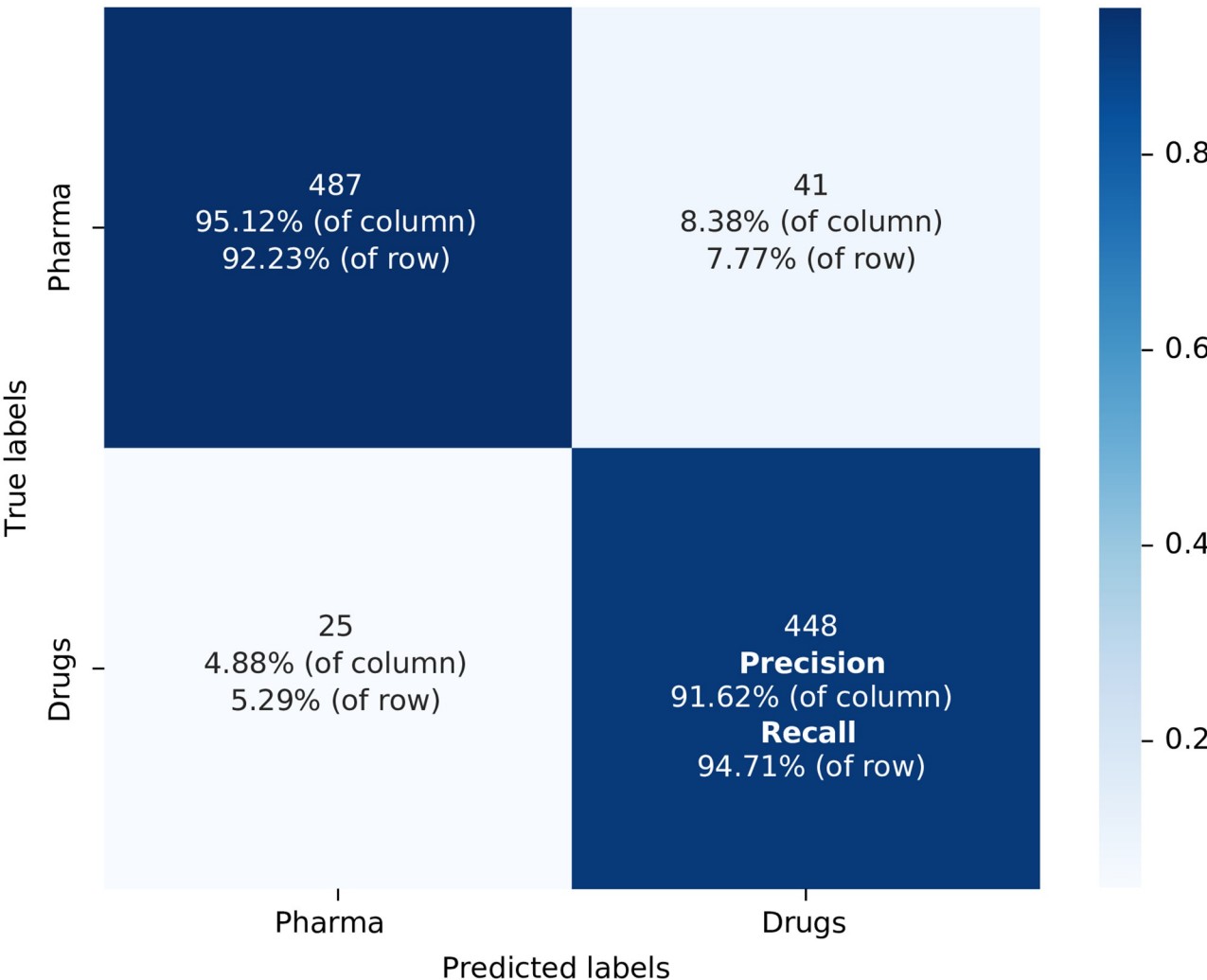

**Fig 3. Confusion matrix for the model used the training dataset built from Orange Book Patents with Biomedical DistilBert as the algorithm model label in Table 1 T: OB; C: G+B.** Source: Authors' elaboration.

dataset comprising 832 patents, which accounts for 0.5% of the total size of the initial training dataset. In this scenario, the accuracy is 91.34% result, the precision 85.88% precision, 92.26% recall, and an F1-score of 0.890. A comparison of the performance metrics of this model with the two preceding ones indicates that the inclusion of family abstracts in the training dataset had a negative impact on its performance. The probable cause of this phenomenon is the introduction of redundant information through the similarities shared among family abstracts which hinders the model's ability to generalize the behaviour of the training dataset to other groups of patents (test dataset) effectively.

In our fourth and final combination, we used Biomedical DistilBERT because of its demonstrated higher efficiency in our previous results. But now we utilised the IQVIA chemical drug patent dataset as the training dataset which, as discussed earlier, it is twice as large as the Orange Book Patents' training dataset we used in the previous two models/scenarios (represented by the steps 9 and 10 in Fig 1). Fig 5 illustrates the confusion matrix for this final model calculated on the testing dataset consisting of 1198 patents (5% of the training dataset). When

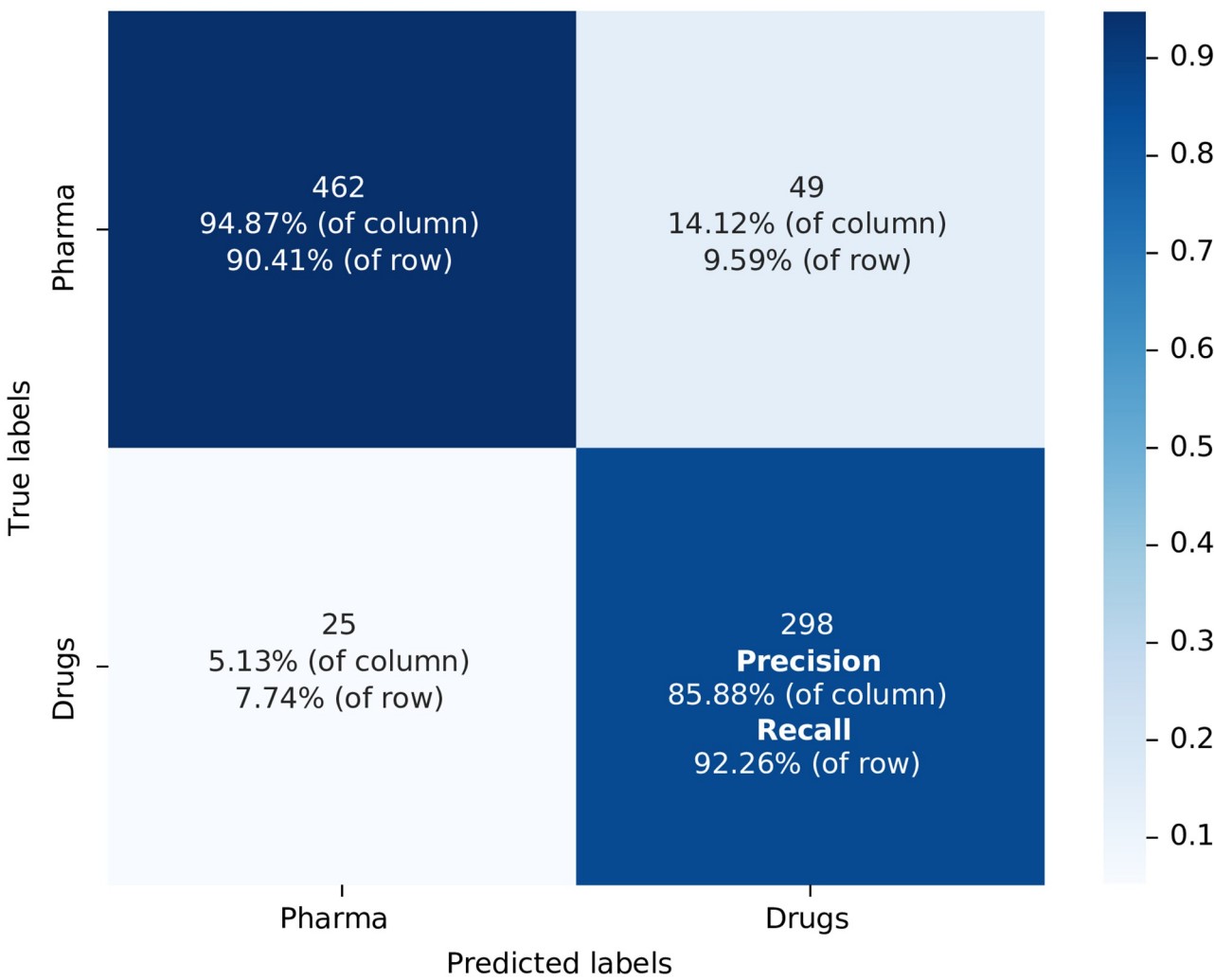

**Fig 4. Confusion matrix for the model used the training dataset built from Orange Book Patents & their Families with Biomedical DistilBert as the algorithm model label in** Table 1 **T: OB+Family; C: G+B.** Source: Authors' elaboration.

calculating accuracy, we obtain a 94.40% result, 91.44% precision, 96.17% recall, and an F1-score of 0.937. Except for a minor decrease in precision, all other performance metrics showed improvement. This indicates that the increased volume of information in the latest training dataset has contributed to the better training of the algorithm, resulting in a subtle enhancement of its overall performance.

To enable the visualization of the classification performed by the most efficient model (the last one) for recognising the two different patent groups using BERT, we began by getting the representation of the embeddings in the BERT pooler layer before applying the last layer of a dense neural network for patent classification. We then projected the embeddings into a two-dimensional space using the Uniform Manifold Approximation and Projection (UMAP), also an AI algorithm. The UMAP algorithm is used to reduce high-dimensional data into a lower-dimensional space while retaining the essential structure and relationships within the data. UMAP accomplishes this by modelling the data's local relationships using nearest-neighbour graphs and optimising a low-dimensional representation that maintains these local

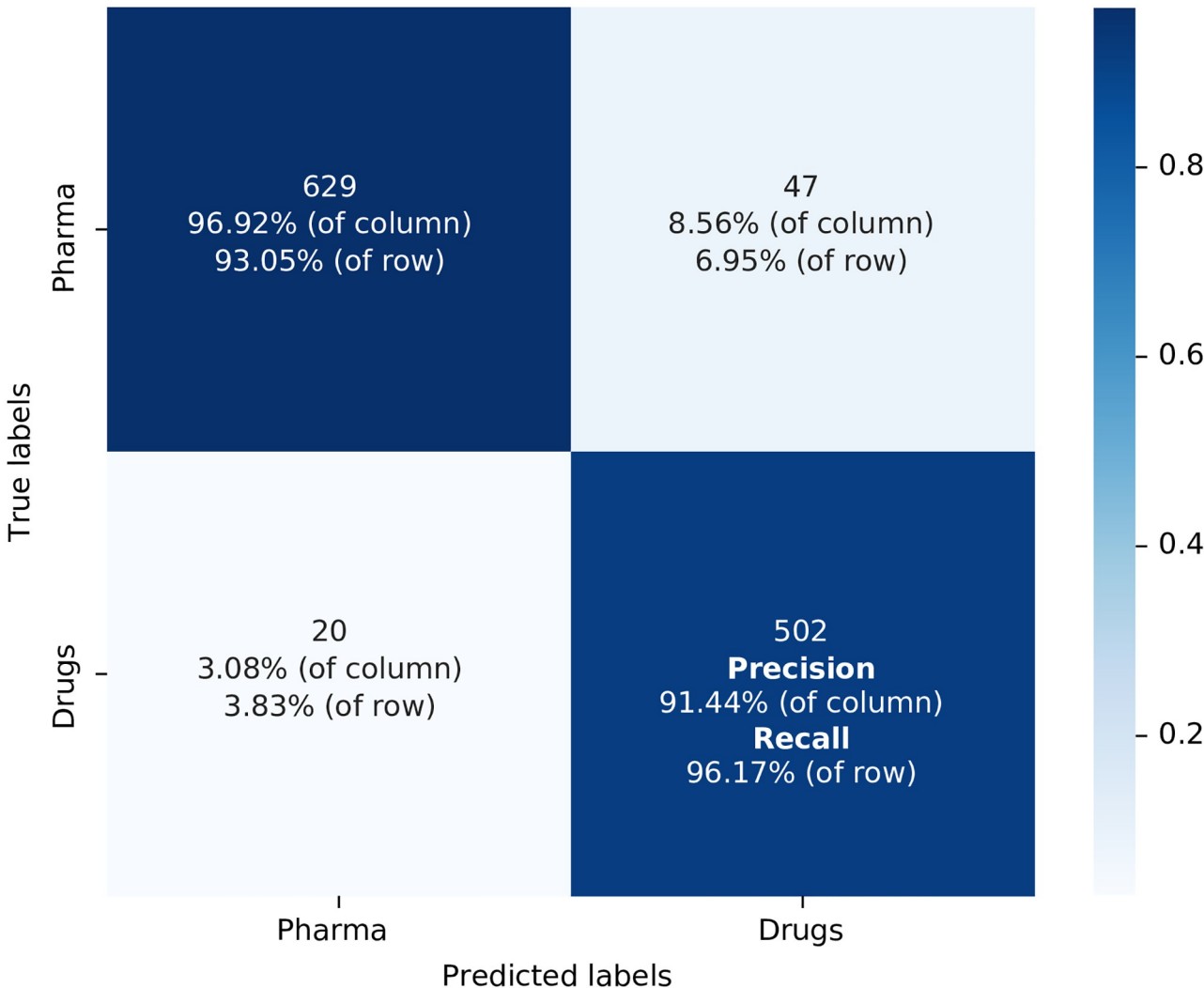

**Fig 5. Confusion Matrix for the model used the training dataset built from IQVIA patents with Biomedical DistilBert as the algorithm model label in Table 1 T: IQVIA; C: G+B.** Source: Authors' elaboration.

relationships as closely as possible. This is achieved through an iterative process of adjusting the positions of the data points in the lower-dimensional space in order to preserve similar relationships and distinguish dissimilar ones. This approach makes UMAP effective for visualisation, clustering, and preserving the global and local structure of the data in a lower-dimensional space. For the projection, we extracted the embedding corresponding to the [CLS] token (the first token in the sequence) for each abstract in the training dataset from the model's pooler. The [CLS] embedding is often used as a representation of the entire sequence, serving as the vector representation of the whole abstract in this case. Subsequently, we applied UMAP to reduce the dimensionality of the space from 768 (the size of the BERT embeddings) to 2 dimensions. Finally, we constructed a scatter-plot in this reduced space, with pharmaceutical patent abstract embeddings represented by the blue circles and drug patents by orange circles. This information we took from the real labels in the training base. In Fig 6, this scatter-plot clearly shows the separation between pharmaceutical and drug patents. Pharmaceutical

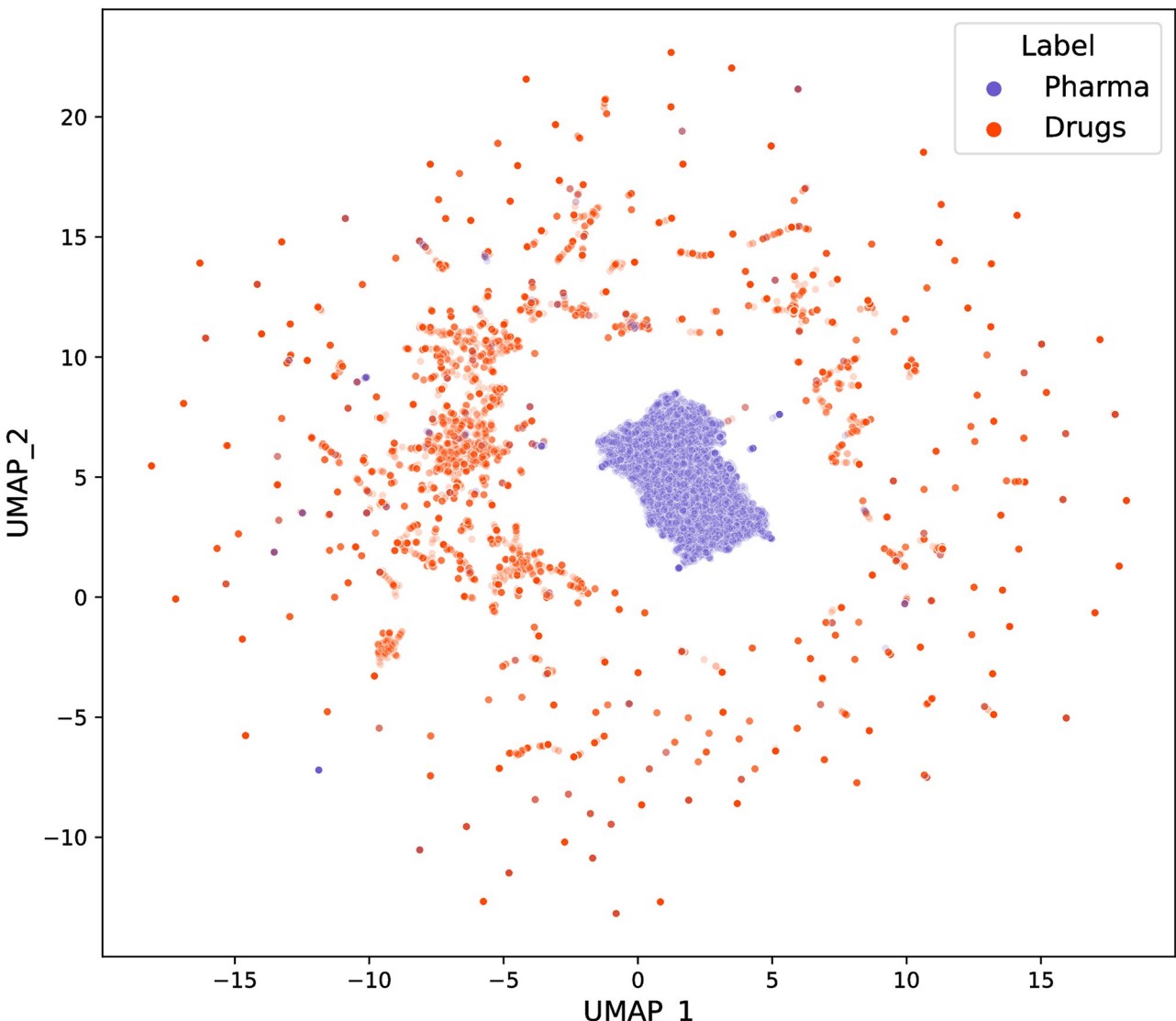

**Fig 6. UMAP projection of BERT embeddings allowing the visualization of group separation.** Source: Authors' elaboration.

patents are concentrated in the centre of the graph, while drug patents are spread around this central group but clearly separated from it. There are very few points associated with drug patents in the region where pharmaceutical patents are concentrated, and there are very few blue dots associated with medicine patents in the region. This indicates that the classification model has achieved high precision.

To enable meaningful comparisons between the performance metrics of various models, Table 1 displays all the relevant data. A notable enhancement in performance occurs when utilising the Biomedical DistilBERT Corpus. Conversely, there is a discernible decline in performance when incorporating the Orange Book patents and their families. The two most effective models deployed the Biomedical DistilBERT Corpus and either the Orange Book Patents or the IQVIA Patents databases for training. We opt for the latter combination because it exhibits a higher accuracy and recall compared to the former one. Moreover, its F1 score is superior, despite having a slightly lower precision score.

**Table 1. Comparing the models' performance metrics.**

| Training Dataset | DistilBERT Corpus | Model label | Accuracy | Precision | Recall | F1-score |
|---|---|---|---|---|---|---|
| Orange Book Patents | General | T: OB; C: G | 91.80% | 89.49% | 93.66% | 0.915 |
| Orange Book Patents | General + Biomedical | T: OB; C: G+B | 93.40% | 91.62% | 94.71% | 0.931 |
| Orange Book & Family Patents | General + Biomedical | T: OB+Family; C: G+B | 91.34% | 85.88% | 92.26% | 0.89 |
| IQVIA Patents | General + Biomedical | T: IQVIA; C: G+B | 94.40% | 91.44% | 96.17% | 0.937 |

Source: Authors' elaboration

## Applying the method to the USPTO and the German patent office

Various applications emerge once the algorithm has been trained to present high performance indices. It may seem counterintuitive to apply the algorithm to the USPTO in the first instance, as we do here, given that the Orange Book already lists granted chemical drug patents that correspond to FDA-approved drugs. There are, however, two reasons why this is appropriate. First, there is the temporal gap between patent grant and drug approval which means that chemical drug patents listed in the Orange Book do not capture the entire universe of chemical drug-related patents in the US. As noted, a patent application related to a therapeutic agent comes early on in the process, typically before a decision to pursue the drug is made and clinical trials undertaken. Usually, only a very small proportion of promising compounds tested invitro and animals enters the clinical trials stage which typically last five to seven years [48]. Until the chemical drug is approved following successful clinical trials, many drug patents would appear in USPTO but not in the Orange Book. Leveraging our algorithm for recently-granted patents at the USPTO would thus capture almost all the universe of granted drug-related patents.

If we add to this temporal gap the time needed for the USPTO to grant the patent in the first place—we calculate this based on USPTO data to be around three years—new potential uses emerge. Among the most intriguing would be applying our model to patents filed at the USPTO in the last decade, for instance, which would identify all the patent applications related to chemical drugs. Although we do not execute it here, longitudinal applications of this kind of proactive identification, combined with Orange Book data, can help identify the likelihood of a chemical drug-related patent application not only being granted but also being linked to an approved drug in the market at a future date, around ten years later (three years needed, on average, for the USPTO to grant the drug-related patent, plus seven years, on average, needed to complete clinical trials and secure the FDA marketing approval). This constitutes the second reason why we turn to the USPTO in the first instance. Research on the pharmaceutical R&D process suggests that only around 10-13% of compounds succeed in clinical trials [48, 49]; many candidates either prove unsuccessful or are discontinued by the drug sponsor for any reason, not least those related to potential financial returns. In short, if the success rate of clinical trials is correct, a considerable share of drug-related patents does not appear in the Orange Book at all.

Using our algorithm with the highest accuracy (labelled IQVIA Patents –Biomedical DistilBert Corpus, above) on a dataset of pharmaceutical patents, i.e. classified under IPC codes A61K, A61P, C07K, and C12N granted by the USPTO during 2010-2020, we identified 150,685 pharmaceutical patents of which 14,348, or approximately 9.5%, are chemical drug-related patents. Some of them would be misclassified, but we know that their number is quite small due to the high-performance metrics detailed in the previous section. This percentage is surprisingly low and, as noted, deserving of further investigation. Chien and colleagues [34]

**Table 2. Top drug patentholders, USPTO 2010-2020.**

|  | Firm Name | Country | Predicted Drug Patents |
|---|---|---|---|
| 1 | NOVARTIS AG | CH | 128 |
| 2 | MERCK SHARP & DOHME | US | 118 |
| 3 | ALLERGAN INC | US | 91 |
| 4 | VERTEX PHARMA | US | 89 |
| 5 | BRISTOL MYERS SQUIBB CO | US | 82 |
| 6 | ABBVIE INC | US | 74 |
| 7 | GILEAD SCIENCES INC | US | 73 |
| 8 | ANTECIP BIOVENTURES II LLC | US | 72 |
| 9 | PURDUE PHARMA LP | US | 70 |

Source: Authors' elaboration

claim that drug patents account for around 4% of total pharmaceutical patents; however, the method used to achieve this result was not explained, making direct comparisons with our finding difficult. Biologic drug-related patents would increase this share, especially as biologics tend to have more patents related to them, although not significantly given that there are relatively fewer biologic than chemical drugs in the market. These types of analyses are beyond the scope of this article; suffice to note that identifying the key patentholders within the drug-related subcategory, as we do in Table 2, would open fruitful avenues of research, including the temporal development of their drug-related patent portfolios.

Because our algorithm was trained on drug-related patent abstracts rather than metadata which is more likely to display local patenting characteristics, in principle it can be used in any patent office in the world. This application would allow the identification of drug-related patents in these offices and identify key patentholders and patenting strategies that potentially reflect more local or regional dynamics. We opted to use our algorithm to identify chemical drug-related patents in the German Patent Office (herein, DPMA), an important pharmaceutical locus in the EU. Two steps were undertaken first. To verify the applicability of our algorithm in patent offices besides the USPTO, we conducted an experiment using a random sample of 90 drug-related patent families, with each family having at least one patent granted by both the USPTO and the European patent Office (EPO). When the algorithm was applied to the abstracts of both the USPTO and EPO patents, it predicted the same classification for 93% of the families, supporting to our hypothesis that it can be effectively applied to patents from other patent offices. The second preparatory step relates to the language used by non-English language patent offices. We leveraged for this purpose Facebook's state-of-the-art M2M-100 model, a non-English centric model that can translate between 100 languages in any direction. Unlike traditional translation models that rely heavily on English-centric data, M2M-100 boasts a massive 7.5 billion sentence dataset spanning 100 languages, ensuring good accuracy across diverse linguistic domains.

The DPMA lists a total of 5,750 pharmaceutical patent applications made between 2010 and 2020. Our algorithm identifies 395 of these as chemical drug-related patents, or approximately 6.9% of total pharmaceutical patent applications. Comparing the main applicants within this group of patents (Table 3) with that we obtain in the case of the USPTO (Table 2) brings to light a noticeable feature: while the USPTO list is dominated by high-profile, global proprietary pharmaceutical companies, the DPMA list includes companies with a seemingly more regional profile alongside prominent ones such as Bayer and Merck. Comparing top drug-related patent applicants identified by the algorithm to those of pharmaceutical patents leads

**Table 3. Top chemical drug-related and top pharma applicants, DPMA 2010-2020.**

|   | Top Drug-related Patent Applicants |  | Predicted Drug Patents |   | Top Pharma Patent Applicants | Pharma Patents |
|---|---|---|---|---|---|---|
| 1 | BEIERSDORF AG | DE | 57 | 1 | HENKEL AG & CO KGAA | 1,808 |
| 2 | BAYER SCHERING PHARMA AG | DE | 31 | 2 | BEIERSDORF AG | 556 |
| 3 | HENKEL AG & CO KGAA | DE | 20 | 3 | MERCK PATENT GMBH | 89 |
| 4 | FRESENIUS MEDICAL CARE DE GMBH | DE | 8 | 4 | EVONIK DEGUSSA GMBH | 80 |
| 5 | ACROVIS BIOSTRUCTURES GMBH | DE | 8 | 5 | BAYER SCHERING PHARMA AG | 77 |
| 6 | RATIOPHARM GMBH | DE | 8 | 6 | FRAUNHOFER GES FORSCHUNG | 72 |
| 7 | STADA ARZNEIMITTEL AG | DE | 7 | 7 | BAYER PHARMA AG | 70 |
| 8 | BAYER PHARMA AG | DE | 6 | 8 | BAYER IP GMBH | 58 |
| 9 | LOHMANN THERAPIE SYST LTS | DE | 6 | 9 | SIEMENS AG | 58 |
| 10 | MERCK PATENT GMBH | DE | 6 | 10 | SCHULZE ZUR WIESCHE ERIK | 41 |
| 11 | AICURIS GMBH & CO KG | DE | 6 | 11 | FORSCHUNGSZENTRUM JUELICH GMBH | 38 |

Source: Authors' elaboration

to a number of interesting insights. First of these is that, both groups are more diverse in the case of the German patent office and not dominated by large proprietary pharmaceutical companies as in the case of the USPTO. The second is that companies that top the list of (general) pharmaceutical patents in the DPMA are not the same as those that top the drug-related patent lists. This suggests that, besides their more regional profile, there is also a specialisation occurring in terms of their predominant orientation. Third, and perhaps more important from a methodological point of view, is that the algorithm is identifying drug-related patents correctly in this case. This is visible in the fact that only around 1% of pharmaceutical patents of Henkel AG & Co, the company with the most patents, are identified as drug-related; conversely, this share is around 10% for Beiersdorf AG, the company with the largest number of drug-related drugs. In short, the algorithm is not simply attributing 6.9% of their patents as belonging to the drug-related patent category.

A final noteworthy feature of the results obtained from applying the algorithm to pharmaceutical patents at the DMPA is that knowledge protected by chemical drug-related patents in Germany is most likely to have developed domestically/regionally rather than elsewhere in the world and then protected in Germany via a priority application. This is because, when we examine the 395 chemical drug-related patents identified by the algorithm, we find that 348 of them (88%) do not have a priority claim, that is, they have been filed at the DMPA first and not associated with previously filed applications in other patent offices. This observation is in line with the more regional profile of the companies that top the list in the case of the DMPA compared to the USPTO. On the other hand, of the 14,348 drug-related patents the algorithm identified at the USPTO, 66, 3% list this office as their first priority. This lower percentage of a much larger body of drug-related patents held by more globally-oriented companies is in line with the US being the most important and dynamic pharmaceutical and patent market in the world.

## Conclusion

We started with the observation of a mismatch between pharmaceutical patents as highly-valuable assets and their justification as a sort of compromise ultimately aimed at enabling new and more effective drugs reaching the market. Besides this, patent disclosure and knowledge diffusion remains an important, even if not the primary, feature of modern patent systems. Nevertheless, as noted, it is difficult to identify and work with specific categories of

pharmaceutical patents beyond the rather crude ones given by patent offices. Focusing on chemical drug-related patents here, and on biologic drug-related patents in a separate forthcoming work, constitutes an attempt to make pharmaceutical patent data less opaque and closer to the disclosure and diffusion ideal of the patent system. Advances in AI, and in NLP in particular, offer us the opportunity to use these knowledge instruments more precisely to address the problem at hand. The ability to train the BERT algorithm to tackle particular tasks is especially useful. We leveraged this characteristic repeatedly by exposing it to additional texts containing terms and structures that are much more likely to be present in the patent subcategory in question, until we obtain the highest possible F1-score, combined with an accuracy of 94.40%. The peculiar nature of the drug R&D and patent grant processes mean that applying the algorithm to the USPTO database of pharmaceutical patents is a worthwhile endeavour. One interesting element that stems from the peculiar nature of these processes is that the algorithm can identify potential chemical drug-related patents up to ten years before drug approval, i.e. at the stage the patent application is made at the USPTO. Applying the algorithm to the German patent office revealed other features, the most relevant of which is perhaps the more regional nature of chemical drug R&D and patenting strategies of the companies involved. While we focused on chemical drug patents, the method we develop can be used to identify any patent group not covered by specific IPC classes. As explained in detail, building one or more training databases with patents previously associated with the target group is a crucial step in training the algorithm. This can then be applied to the entire patent database to generalise the classification 'learned' during training. Enabling further uses is one reason why we have explained our steps in detail. More generally, we have been careful throughout to make our 'craftsmanship' and imprint on the algorithm's structure and training, and therefore on its outcomes. The detailed explanation of our method constitutes an effort to keep AI tools open and to allow their scrutiny and evaluation by the broader community.

## Acknowledgments

The authors want to express their sincere gratitude to Professor Ken Shadlen (LSE) for help with this work.

## Author Contributions

**Conceptualization:** Leonardo Costa Ribeiro.

**Data curation:** Leonardo Costa Ribeiro.

**Formal analysis:** Leonardo Costa Ribeiro, Valbona Muzaka.

**Funding acquisition:** Leonardo Costa Ribeiro.

**Investigation:** Leonardo Costa Ribeiro, Valbona Muzaka.

**Methodology:** Leonardo Costa Ribeiro.

**Software:** Leonardo Costa Ribeiro.

**Supervision:** Leonardo Costa Ribeiro.

**Validation:** Leonardo Costa Ribeiro.

**Visualization:** Leonardo Costa Ribeiro.

**Writing – original draft:** Leonardo Costa Ribeiro, Valbona Muzaka.

**Writing – review & editing:** Leonardo Costa Ribeiro, Valbona Muzaka.

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
