## [Decision Letter · Decision Letter 0]

9 Jul 2024

PONE-D-24-11668Needle in a haystack: Harnessing AI in drug patent searches and predictionPLOS ONE

Dear Dr. Muzaka,

Thank you for submitting your manuscript to PLOS ONE. After careful consideration, we feel that it has merit but does not fully meet PLOS ONE’s publication criteria as it currently stands. Therefore, we invite you to submit a revised version of the manuscript that addresses the points raised by the reviewers.

We look forward to receiving your revised manuscript.

Kind regards,

Pradeep Kumar, Ph.D.

Academic Editor

PLOS ONE

Reviewers' comments:

Reviewer's Responses to Questions

**Comments to the Author**

1. Is the manuscript technically sound, and do the data support the conclusions?

Reviewer #1: Partly

Reviewer #2: Yes

Reviewer #3: Yes

2. Has the statistical analysis been performed appropriately and rigorously? 

Reviewer #1: Yes

Reviewer #2: Yes

Reviewer #3: Yes

3. Have the authors made all data underlying the findings in their manuscript fully available?

Reviewer #1: No

Reviewer #2: Yes

Reviewer #3: No

4. Is the manuscript presented in an intelligible fashion and written in standard English?

Reviewer #1: Yes

Reviewer #2: Yes

Reviewer #3: Yes

5. Review Comments to the Author

Reviewer #1: The authors propose the use of distilBERT for pharmaceutical patent category classification using the Biomedical Corpus, Orange book Patents and IQVIA datasets. The topic is relavant and the context it is applied in is unique, but it lacks technical robustness required for reproduction and to gain relevant insights that would allow us to determine how well it generalises. There are adjacent works that are not outlined such as:

Lee, C., Kwon, O., Kim, M., & Kwon, D. (2018). Early identification of emerging technologies: A machine learning approach using multiple patent indicators. Technological Forecasting and Social Change, 127, 291-303.

Balsmeier, B., Assaf, M., Chesebro, T., Fierro, G., Johnson, K., Johnson, S., ... & Fleming, L. (2018). Machine learning and natural language processing on the patent corpus: Data, tools, and new measures. Journal of Economics & Management Strategy, 27(3), 535-553.

There are also technical faults that need to be addressed (see annotated file) and more clarity is required in why certain decisions were made in the method design. I also think it would valuable if you added a t-SNE plot of the embeddings to show the quality of category groupings and a precision-recall to show how well define the decision boundary is of predictions.

Reviewer #2: The patent documents consist of many important information which is not accessible easily. In the current manuscript author non proposed an AI based method to analyze the patent databases.

1. Abstract: The abstract is not cohesive, can be revised to overall summarize the outcome of the analyses.

2. Introduction: This section is unnecessarily long. For ex. The TRIPS section (page 3, last para) seems irrelevant and can be removed. The other paragraphs can also be squeezed.

3. Figure 1, pg 11: The figure legend needs to be self-explanatory so that readers get an idea of what has been done without reading the methods section. Also the text “Source: Authors’ elaboration can be removed. ”

4. The manuscript is no as per PlosOne format. i.e. footnotes not encouraged, references in Vancouver style etc. This needs to be considered.

Reviewer #3: The submitted manuscript provides for the development and evaluation of an Artificial Intelligence tool for the searching and prediction of drug patents. The authors have written a very interesting paper and have meticulously provided a background to the current issues facing patent searches for drug molecules and have provided a comprehensive breakdown of how the AI tool was developed and how the study has been performed. There are however a few amendments for the authors to address prior to this paper being suitable for publication. A list of these are as follows:

1. There are currently a number of widely available AI tools for patent searching. A discussion around these would benefit this article by providing the advancement of the AI tool developed in this study compared to other available AI tools for patent searching.

2. A brief discussion should be provided on the versatility of the developed AI tool and its potential to be used for patents outside of the pharmaceutical/drug space in the future.

3. Any future advancement of the developed AI tool that could improve its functionality should also be highlighted.

6. PLOS authors have the option to publish the peer review history of their article (what does this mean?). If published, this will include your full peer review and any attached files.

Reviewer #1: **Yes**

Reviewer #2: No

Reviewer #3: No

---

## [Author Response · Author response to Decision Letter 0]

11 Sep 2024

Please see our Response to Reviewers document (we include in that document our response to points raised by the editors).

---

## [Editor Report · Decision Letter 1]

17 Sep 2024

Needle in a haystack: Harnessing AI in drug patent searches and prediction

PONE-D-24-11668R1

Dear Dr. Muzaka,

We’re pleased to inform you that your manuscript has been judged scientifically suitable for publication and will be formally accepted for publication once it meets all outstanding technical requirements.

Kind regards,

Pradeep Kumar, Ph.D.

Academic Editor

PLOS ONE
---

## [Editor Report · Acceptance letter]

23 Sep 2024

PONE-D-24-11668R1 

PLOS ONE

Dear Dr. Muzaka, 

I'm pleased to inform you that your manuscript has been deemed suitable for publication in PLOS ONE. Congratulations! Your manuscript is now being handed over to our production team.

Kind regards, 

on behalf of

Prof. Pradeep Kumar 

Academic Editor

PLOS ONE